# What Is Machine Learning, Artificial Neural Networks and Deep Learning?—Examples of Practical Applications in Medicine

**DOI:** 10.3390/diagnostics13152582

**Published:** 2023-08-03

**Authors:** Jakub Kufel, Katarzyna Bargieł-Łączek, Szymon Kocot, Maciej Koźlik, Wiktoria Bartnikowska, Michał Janik, Łukasz Czogalik, Piotr Dudek, Mikołaj Magiera, Anna Lis, Iga Paszkiewicz, Zbigniew Nawrat, Maciej Cebula, Katarzyna Gruszczyńska

**Affiliations:** 1Department of Biophysics, Faculty of Medical Sciences in Zabrze, Medical University of Silesia, 41-808 Zabrze, Poland; znawrat@sum.edu.pl; 2Paediatric Radiology Students’ Scientific Association at the Division of Diagnostic Imaging, Department of Radiology and Nuclear Medicine, Faculty of Medical Science in Katowice, Medical University of Silesia, 40-752 Katowice, Poland; katarzyna.bargiel@op.pl (K.B.-Ł.); wiktoriabart96@gmail.com (W.B.); 3Bright Coders’ Factory, Technologiczna 2, 45-839 Opole, Poland; 4Division of Cardiology and Structural Heart Disease, Medical University of Silesia, 40-635 Katowice, Poland; kozlik.maciej@gmail.com; 5Student Scientific Association Named after Professor Zbigniew Religa at the Department of Biophysics, Faculty of Medical Sciences in Zabrze, Medical University of Silesia, Jordana 19, 41-808 Zabrze, Poland; michal.janik0801@gmail.com (M.J.); lukczog@gmail.com (Ł.C.); piotrekd233@gmail.com (P.D.); mikaczu7422@gmail.com (M.M.); igapaszkiewicz.ip@gmail.com (I.P.); 6Cardiology Students’ Scientific Association at the III Department of Cardiology, Faculty of Medical Sciences in Katowice, Medical University of Silesia, 40-635 Katowice, Poland; lis.anna9898@gmail.com; 7Individual Specialist Medical Practice Maciej Cebula, 40-754 Katowice, Poland; maciejmichalcebula@gmail.com; 8Department of Radiodiagnostics, Invasive Radiology and Nuclear Medicine, Department of Radiology and Nuclear Medicine, School of Medicine in Katowice, Medical University of Silesia, Medyków 14, 40-752 Katowice, Poland; kgruszczynska@sum.edu.pl

**Keywords:** AI, artificial intelligence, medicine, AI in medicine

## Abstract

Machine learning (ML), artificial neural networks (ANNs), and deep learning (DL) are all topics that fall under the heading of artificial intelligence (AI) and have gained popularity in recent years. ML involves the application of algorithms to automate decision-making processes using models that have not been manually programmed but have been trained on data. ANNs that are a part of ML aim to simulate the structure and function of the human brain. DL, on the other hand, uses multiple layers of interconnected neurons. This enables the processing and analysis of large and complex databases. In medicine, these techniques are being introduced to improve the speed and efficiency of disease diagnosis and treatment. Each of the AI techniques presented in the paper is supported with an example of a possible medical application. Given the rapid development of technology, the use of AI in medicine shows promising results in the context of patient care. It is particularly important to keep a close eye on this issue and conduct further research in order to fully explore the potential of ML, ANNs, and DL, and bring further applications into clinical use in the future.

## 1. Introduction

Artificial intelligence (AI) has gained significant attention over the past decade, with machine learning (ML) and deep learning (DL) being popular topics [1,2]. While these terms are related, they have distinct meanings and cannot be used interchangeably. This article aimed to provide clinicians with accessible information on the AI methods used in medicine without relying heavily on technical jargon. Various definitions of AI exist, with some focusing on intelligent systems making decisions to achieve goals, while others emphasise machines mimicking cognitive functions, like learning and problem solving. ML and DL are specific methods within the broader field of AI, contributing to the automation of intellectual tasks performed by humans. This article will also cover FDA-approved AI solutions in medicine, discussing the machine learning methods that have been employed, the type of data integrated, their overall performances, as well as their advantages and disadvantages. The content of this study will be presented in a manner understandable to those without technical expertise in the field.

## 2. Glossary

**Model architecture**—describes how the data is processed, transmitted, and analysed within the machine learning algorithm, which influences its efficiency and effectiveness in solving problems.**Data exploration**—the process of analysing and summarising a large dataset to gain insight into the relationships and patterns that exist within the data.**Binary classification**—a type of classification in which the aim is to assign one of two possible classes (labels) to an object: positive or negative, true or false, etc.**Logistic function**—a sigmoidal mathematical function that transforms values from minus infinity into plus infinity to the range (0, 1), allowing not only non-linearity, but also probability, e.g., binary classification.**Input variables**—also known as independent variables, explanatory variables, predictor variables, etc., and are variables that are used to describe or explain the behaviour, trends, or decisions of the target variables.**Target variables**—also known as dependent variables, outcome variables, etc., and are variables that are studied or predicted in statistical analysis and machine learning. Target variables are dependent on and are described using explanatory variables.**Bayes theory**—used to calculate the probability of an event, having prior information about that event.**Sentiment analysis**—the process of automatically determining emotions, opinions, and moods expressed in the text. This can be in the form of product reviews, comments on online forums, tweets on Twitter, or other forms of textual communication. The purpose of sentiment analysis is to gain an automatic understanding of whether a text is positive, negative, or neutral.**Training**—the process of formatting a model to interpret the data to perform a specific task with a specific accuracy. In this case, it is the determination of the hyperplane.**Hyperplane**—a set having *n* − 1 dimension, relative to the *n*-dimensional space in which it is contained (for *n* = two-dimensional space it has one dimension (point); for *n* = three-dimensional space it has two dimensions (line)).**Training objects**—a set of objects used to determine the hyperplane with the model.**Support vectors**—at least two objects at the shortest distance from the hyperplane belonging to two classes.**Class**—a group, described on numerical ranges, to which an object can be assigned—i.e., classified.**Cluster**—a hyperplane-limited space in a data system in which the presence of an object determines the class assignment.**Neuron**—the basic element of a neural network, which connects to other neurons through transmitting data to each other.**Weight**—the characteristic that the network designer provides to the connections between the neurons to achieve the desired results.**Recursion**—referring a function to the same function using the network being trained.**Layer**—a portion of the total network architecture that is differentiated from the other parts due to having a distinctive function.**Activation function**—takes the input from one or more neurons and maps it to an output value, which is then passed onto the next layer of neurons in the network.**Hidden layer**—in an artificial neural network, this is defined as the layer between the input and output layers, where the result of their action cannot be directly observed.**Input layer**—layer where the data are collected and passed onto the next layer.**Output layer**—layer which gathers the conclusions.**Backpropagation**—sending signals in the reverse order to calculate the error associated with each individual neuron in an effort to improve the model.**Cost function**—a function that represents the errors occurring in the training process in the form of a number. It is used for subsequent optimisation.**Receptive field**—a section of the image that is individually analysed using the filter.**Filter**—a set of numbers that are used to perform computational operations on the input data on splices. It is used to extract features (e.g., the presence of edges or curvature).**Convolution**—integral of the product of the two functions after one is reflected about the y-axis and shifted.**Pooling**—reducing the amount of data representing a given area of the image.**Matrix**—a mathematical concept; a set of numbers which is used, among other things, to recalculate the data obtained from neurons.**Skip connections**—a technique used in neural networks that allows information to be passed from one layer of the network to another, while skipping intermediate layers.

## 3. Data and Its Relevance to Neural Networks

Data are essential for any data analysis method. It consists of facts or information in different forms, like sound, text, values, and images. In AI, data are used to train models and enable them to make decisions and perform tasks, like classification or text generation. The proper interpretation of data is crucial for training neural network models. Data can be structured (easily organised) or unstructured (difficult to classify). Structured data, such as phone numbers or financial data, can be processed and analysed effectively. Unstructured data are those that are not easily classified and organised; examples include images and sound files [3].

The input data mentioned above needs to be normalised for the successful learning process of a neural network. Meanwhile, the output data from the neural network are used for predictions or solutions.

To determine the training dataset size for a machine learning model, several parameters are required to be defined, which include the complexity of the model, the complexity of the learning algorithm, the need for labelling, the definition of an acceptable margin of error, and the diversity of the inputs used [4]. In short, the input data must be prepared in terms of the task and the expected results.

Data processing can be approached in two ways: focusing on the model or focusing on the data. The first approach aims to enhance the performance of the machine learning (ML) model by selecting an efficient model architecture and learning process. It involves improving the model’s code without changing the amount of data. This method is gaining increasing levels of popularity as it eliminates the need for extensive data collection. The second strategy emphasises working on the data itself, and involves modifying and enhancing the datasets to improve the accuracy of the decisions being made [5].

## 4. Machine Learning Models

Machine learning is a subset of AI that involves building computer models that are capable of learning and making independent predictions or decisions based on the provided data. These models continually improve their accuracy through learned data.

The author of the name machine learning, Arthur Samuel, used this phrase in 1959 to describe “the ability of computers to learn without programming new skills directly” [6]. Using the available datasets, a machine learning algorithm, supported with a mathematical model, generates predictions or specific decisions. The main types of machine learning are supervised and unsupervised learning.

### 4.1. Supervised Learning

Supervised machine learning assumes that the model has been trained on a similar dataset to the problem at hand, consisting of the input data and the corresponding output data. Once the model learns the relationship between the input and the output, it can classify new unknown datasets and make predictions or decisions based on them. This type of learning is divided into two methods: classification and regression.

In supervised machine learning, e.g., a photograph that the algorithm classifies as either a cat or a dog, representing a two-class classification problem, the solutions are typically binary, in the form of either yes or no. Another example is handwriting recognition, where the software matches the handwritten characters (output data) to their corresponding printed counterparts (classes).

Regression is a fundamental type of supervised learning that predicts continuous values using input data. For instance, in healthcare, regression can be applied to forecast the medical costs. The input data would include drug prices, the required medical equipment, and staff expenses, while the output would be the total treatment cost. Through training these models with the input and output data, predictions for the total treatment cost can be made for new inputs.

### 4.2. Unsupervised Learning

Unsupervised ML differs from supervised ML in its use of unannotated data, which has not been previously labelled by humans nor algorithms. The model learns from input data without expected values, and the available dataset does not provide answers to the given task. Instead of labelling or predicting outputs, this algorithm focuses on grouping the data based on their characteristics. The goal is to teach the machine to detect patterns and group the data without a single correct answer. There are two types of unsupervised learning: clustering and association.

Clustering involves grouping the data based on their similarities and differences. For example, animals can be divided into groups based on their visual features determined using the model.

Association is a method of analysing the relationships between data in a dataset. For instance, the algorithm can pair people buying mattresses for pressure sores with those ordering products to aid in the healing of pressure sores. This method is commonly used in marketing strategies.

## 5. Classical Methods of Machine Learning

### 5.1. k-Nearest Neighbour Algorithms

The k-nearest neighbour (kNN) method is a popular method used in data mining and statistics. The kNN method is a type of algorithm that predicts the correct class of the test data by calculating the distance between the test data and all the training points (Figure 1). It then shows the number of k (training) points that are close to the test data. In the case of regression, the obtained value is the average of the selected training points, “k” [7].

Hamed et al. adapted the kNN algorithm for handling incomplete COVID-19 datasets. They implemented a novel variant called kNNV and tested it on an Italian dataset from Italian Society of Medical and Interventional Radiology (SIRM). Their results demonstrated an effective and accurate COVID-19 case classification (with an accuracy of 0.72–1, and a precision of 0.7–1) [8]. Bellino et al. implemented the kNN algorithm for automatic classification in the operating theatre, aiding neurophysiologists and neurosurgeons in deep brain stimulation electrode fixation for Parkinson’s disease patients. This system, when refined, will reduce surgery time and improve effectiveness [9].

### 5.2. Linear Regression Algorithms

Linear regression predicts the value of a dependent variable from an independent variable (an example of which has been displayed in Figure 2). It generates a simple, interpretable formula for predictions and is widely used in Python and Excel, as well as across a range of fields, like science, biology, business, and behavioural science [10].

An example of the use of these linear regression algorithms is the study presented by Garcia et al. on estimating the carotid-to-femoral pulse wave velocity (cf-PWV) using multiple linear regression (MLR). This model used blood pressure and/or photoplethysmography. Their results suggest that ML (specifically MLR) combined with a semi-classical signal analysis method could be a valuable automated tool for efficient cf-PWV assessments in the future [11].

### 5.3. Logistic Regression Algorithms

Logistic regression predicts the probability of an object belonging to one of two classes. It uses a logistic function to transform predicted values between zero and one. This algorithm establishes the relationship between the input variables and the target variable. Logistic regression has found its application in market research, medical analysis, and banking to understand variable relationships and predict their likely outcomes, such as illness or customer purchases. It helps in answering the questions concerning the likelihood of events or group membership based on the characteristics of the data, like age and education [12,13,14,15].

### 5.4. Naive Bayes Classifier Algorithms

Sentiment analysis automatically identifies emotions, opinions, and moods in the text, such as product reviews, forum comments, and “tweets”. It determines whether the text is positive, negative, or neutral. The Naive Bayes classifier assumes feature independence and quickly categorises information based on this assumption (Figure 3). It is widely used in spam filtering, text classification, and sentiment analysis [16,17,18].

### 5.5. Support Vector Machines (SVMs)

Support vector machines (SVMs) are classed as another type of machine learning model that can be used to classify objects. SVMs involve finding the hyperplane that best separates objects from two different classes (Figure 4). However, in order to allocate these objects to more than one class, separate binary classification training is required [19].

In an n-dimensional data system (*n* ≥ 1), objects are mapped based on their properties. A machine determines a separating hyperplane that maximises the margin between two classes of training objects [20]. These objects are referred to as support vectors [21]. The hyperplane can be a point, line, or plane depending on the object’s dimensions. Its purpose is to define the class membership by assessing which side of the boundary an object falls on. In some cases, an object may possess the properties of another class, leading to the formation of a soft margin, where objects can be located outside the defined cluster without affecting the hyperplane position. If a single hyperplane cannot separate two data clusters, a kernel function can be used to transform the data and achieve a linear separation of the classes [22].

SVMs are valued for their ability to handle non-linear data and high-dimensional classification tasks, making it suitable for processing large datasets [23]. It excels at recognising intricate patterns in complex datasets and finds applications across diverse areas, like handwriting recognition, credit card fraud detection, and facial recognition [24].

In a study by Zhang et al., SVMs were used to diagnose diabetes using tongue photographs. Tongue images were captured from 296 diabetic patients and 531 non-diabetic patients using a TDA-1 camera, respectively. These researchers focused on colour and texture analysis of the tongue’s body and surface. SVM models were trained using this data, resulting in an algorithm that achieved a superior efficiency of 79.72% in diabetes detection from tongue images compared to existing methods [25].

### 5.6. AdaBoost

Adaboost, short for adaptive boosting, is a machine learning algorithm used for binary classification and regression tasks. It is a relatively new non-linear machine learning algorithm [26]. AdaBoost is a boosting technique that iteratively trains multiple weak models on different subsets of the training data and assigns higher weights to the misclassified instances in each iteration (as shown in Figure 5). In subsequent iterations, the algorithm focuses more on the misclassified samples, allowing the weak models to learn from their mistakes and improve on their performance, following which they are combined to form a single strong classifier [27]. 

Adaboost is widely used in computer-aided diagnosis (CAD) and can support medical practitioners to make critical decisions regarding their patients’ disease conditions, such as Alzheimer’s disease, diabetes, hypertension, and various cancers [28]. Furthermore, it can help to improve the accuracy in classifying diseases, predicting patient outcomes, and detecting abnormalities in medical images [29,30].

**Figure 5 diagnostics-13-02582-f005:**
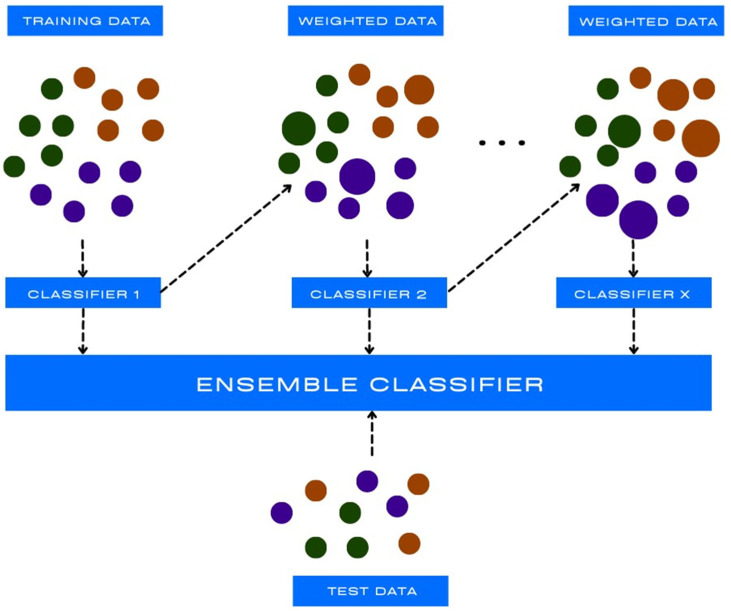
Diagram of the AdaBoost algorithm—different size circles stand for samples with more associated weights, various colors of the circles stand for different subsets of data [31].

### 5.7. XGBoost

XGBoost, which stands for extreme gradient boosting, is a powerful machine learning algorithm that has gained significant popularity across various domains, including in medicine. It is an ensemble learning method that aggregates the predictions of many individually trained weak decision trees to create a more accurate and more powerful model [32].

In medicine, XGBoost has been successfully applied to a wide range of tasks, such as disease diagnosis, prognosis, treatment selection, and patient outcome prediction [33,34,35]. One of the key strengths of the XGBoost algorithm is its ability to handle diverse types of medical data, including structured data (e.g., patient demographics and laboratory results) and unstructured data (e.g., medical images and clinical notes).

## 6. Neural Networks

Artificial neural networks resemble the human brain (Figure 6) and comprise multiple perceptrons or ‘neurons’ that process and transmit information. In understanding their functioning (Figure 7), we begin with inputting data, such as images, text, or sound. This data traverses the network, processed using successive layers of perceptrons until reaching the output. Each layer contains multiple neurons that process the input data.

Neural networks are widely used in image recognition, natural language processing, speech recognition, and stock price prediction. They consist of interconnected neurons and come in various types, as listed below:Perceptron networks: simplest neural networks with an input and output layer composed of perceptrons. Perceptrons assign a value of one or zero based on the activation threshold, dividing the set into two.Layered networks (feed forward): multiple layers of interconnected neurons where the outputs of the previous layer neurons serve as the inputs for the next layer. The neurons of each successive layer always have a +1 input from the previous layer. Enables the classification of non-binary sets, and are used in image, text, and speech recognition.Recurrent networks: neural networks with feedback loops where the output signals feed back into the input neurons. Can generate sequences of phenomena and signals until the output stabilises. Used for sentiment analysis and text generation.Convolutional networks, also known as braided networks, are described in the next paragraph.Gated recurrent unit (GRU) and long short-term memory (LSTM) networks: perform recursive tasks with the output dependent on previous calculations. They have network memory, allowing them to remember data states across different time steps. These networks have longer training times and are applied in time series analysis (e.g., stock prices), autonomous car trajectory prediction, text-to-speech conversion, and language translation.

In order to train the perceptrons, weights are adjusted to minimise the difference between the output and the expected signal. The network also learns through the greatest gradient decline method, adjusting the step lengths in the opposite direction. If the target value at a new point surpasses the starting point, the steps are reduced until the desired value has been achieved (Figure 8).

Backpropagation is another type of ML method that calculates the error for neurons in the last layer and propagates it backwards to the earlier layers. This efficient algorithm has been widely used in research. This network can be tested with new data to assess its performance in recognizing previously unseen information.

### Vision Transformers

As transformers have become more and more successful in solving NLP tasks, they can be used as powerful alternatives to CNNs [36]. Vision transformers have the input image split into rectangular patches, and with the position index combined, they are processed with the standard transformer encoder block without requiring any convolutional layers and reduce the risk of adversarial attacks as a result [37].

## 7. Deep Learning

### 7.1. Differences between the DNNs and ANNs

Based on the number of neural networks, we distinguished between ANNs, which are neural networks consisting of a single hidden layer, and DNNs, which are neural networks that have multiple hidden layers (as shown in Figure 9). This allows the network to understand and mimic more complex and abstract behaviours [38].

### 7.2. Deep Neural Network (DNN) Classifiers

The DNN is a machine learning method with multiple hidden layers. It processes information from the input to the output, utilising weights and backpropagation to minimise the formation of errors. Adding more hidden layers improves the results obtained with DNNs, but also increases their computational and memory requirements as a consequence [40,41,42,43].

In the study published by Han et al., the ability of the ANNs and DNNs in evaluating drug formulations was assessed using 145 formulations with their reported dissolution times. Various factors, including the drug components, filler amount, and manufacturing parameters, were all taken into account. Both the ANNs and DNNs achieved a predictive value of 85.6% in the training set. The validation sets showed an 80% efficacy for ANNs and 85% for DNNs, respectively. However, on the crucial test set, DNNs significantly outperformed ANNs with an effectiveness of 85% compared to 80%, respectively. These results suggest that DNNs are better at predicting unknown data than ANNs [44].

Deep neural networks can be found in numerous healthcare sectors due to their effectiveness. They are currently applied from medical imaging, diagnosis, drug development, prognosis, and risk assessment, to remote monitoring and sports medicine [45,46,47]. The largest number of recent studies report the use of DNNs in the analysis of radiological images, among which include: models detecting apparent and non-apparent scaphoid fractures using only plain wrist radiographs, algorithms for COVID-19 detection from CXR images, models for mammography screening, and tools for the segmentation of intracerebral haemorrhage on CT scans [48,49,50,51]. DNNs are also involved in assessing endomyocardial biopsy data of patients with myocardial injury [52]. Others can identify hypertension on the basis of ballistocardiogram signals or aortic stenosis using audio files [53,54].

Models designed to make diagnoses based on the provided data are gaining popularity—these networks can learn patterns and extract valuable insights from large datasets, assisting in the identification of diseases, like cancer, diabetic retinopathy, and cardiovascular diseases [55,56,57].

### 7.3. Convolutional Neural Networks

Convolutional neural networks (CNNs) consist of input, spline, auxiliary, and output layers. They detect visual patterns obtained from raw image pixels using hidden layers. CNN models analyse receptive fields with filters [58]. Non-linear functions extract information about the image features. ‘Pooling’ reduces data and speeds up computation. This allows for the finding of similar features across the image for pattern analysis [59].

A study by Chamberlin et al. compared the performance of a CNN with expert radiologists in detecting pulmonary nodules and in determining the coronary artery calcium volume (CACV) on low-dose chest CT (LDCT) images. The study included 117 patients for pulmonary nodules and 96 for CACV, respectively. AI results demonstrated an excellent concordance with the radiologists (CACV ICC = 0.904, Cohen’s kappa for pulmonary nodules = 0.846) along with a high sensitivity/specificity (CACV: sensitivity = 0.929, specificity = 0.960; pulmonary nodules: sensitivity = 1, specificity = 0.708, respectively) [60].

Considering CNNs as a form of large-scale learning, it is therefore crucial to understand how to integrate the knowledge acquired from all datasets without putting too much effort into it. However, the performance of CNNs can be impacted through various factors, the most important of which include the selection of activation functions and the quantity of hidden layers. As a result, the accuracy of each CNN experiment fluctuates based on the sizes of the hidden layers that are chosen [61].

### 7.4. Auto-Encoders (Unsupervised)

The auto-encoder (AE) neural network is an unsupervised learning model that reconstructs input images. It utilises an encoder to compress the data and a decoder to reconstruct the output layer (Figure 10). Auto-encoders are effective in data compression and visualisation, particularly in the initial learning phase of neural networks. They aid in identifying patterns in data series. Initial training with auto-encoders addresses potential issues, such as excessive parameters in the DNNs or disparities in gradient magnitudes between the higher and lower layers [62,63].

Auto-encoders are commonly applied in clustering, which involves grouping similar data together, particularly in large datasets. They can identify patterns in data arrangements, such as the sequence of natural numbers or the pattern of a geometric sequence.

Auto-encoders have been used for anomaly detection in brain MRIs. Baur et al. utilised pre-trained auto-encoders to model the normal white matter anatomy of the brain. By reconstructing images and comparing them to the standard model, anomalies and errors could be identified as outliers. This approach is beneficial for detecting uncommon pathologies that may be missed from supervised learning. However, none of the employed autoencoder-based models were able to accurately restore the pathological equivalents of the baseline samples [65].

### 7.5. Segmenting Neural Networks (e.g., UNET and Lung Segmentation)

Current medical image segmentation methods often rely on full CNNs with a U-shaped structure, such as UNET networks. UNET networks feature a symmetric encoder–decoder architecture with minimal connections. The encoder extracts deep features with large receptive fields through convolutional and down-sampling layers. The decoder then up-samples these features to match the input resolution, enabling pixel-level semantic prediction. Minimal connections are primarily used to combine high-resolution features and different scales at the end of the process, reducing the extent of data loss from down-sampling [66].

UNET networks have various applications, including face recognition in images, which are commonly used in smartphones. Shamim et al. proposed convUnet, a modified UNET architecture with additional convolutional layers in each decoder block to identify matte glass areas in lung computed tomography (CT) images. This enhancement significantly improved the segmentation performance for interconnected lung areas in CT imaging. This method shows promise for rapid COVID-19 diagnosis and quantification of infected lung regions [67].

### 7.6. Generative Adversarial Networks

A generative adversarial network (GAN) consists of a generator and a discriminator. GANs learn to generate realistic data by competing against each other. They have been successfully applied in medicine for generating various types of medical images, such as mammograms, CT scans, and MRIs. This allows for the training of models that require diverse image data. GANs are also utilised in generating medical data, including electrocardiograms (ECGs), which can effectively train other models. 

### 7.7. Transfer Learning

Transfer learning is a machine learning technique that aims to enhance the performance of a target task by leveraging the knowledge or representations learned from a different but related source task (Figure 11). It involves the transfer of knowledge, skills, or features from a pre-trained model (source domain) to a new task (target domain) with limited labelled data [68]. However, transfer learning is not without its limitations and potential drawbacks. One notable limitation is the assumption that the source and target domains share some degree of similarity or relatedness. Furthermore, if the domains differ significantly, the transferred knowledge may not be relevant nor applicable to the target domain, resulting in a degraded performance. In such cases, the benefits of transfer learning diminish, and a domain-specific or task-specific model may be more appropriate [69].

### 7.8. Few-Shot Learning

Few-shot learning is a machine learning paradigm that deals with the task of learning new concepts or categories from only a few labelled examples. It addresses the challenge of training accurate models with limited labelled data, which is a common scenario in many real-world applications. Unlike traditional machine learning approaches that require large amounts of labelled data for each class, few-shot learning aims to generalise the knowledge obtained from a small support set to classify or generate instances from novel classes [70]. One significant challenge is the scarcity of labelled examples, which makes it difficult for models to capture the underlying patterns and variances in the data (Figure 12). This can lead to a limited generalization capability along with a poor performance on novel classes or instances which do not present in the support set. Another potential defect is the vulnerability of few-shot learning models to overfitting. With limited labelled data, these models may be more prone to fitting noise or idiosyncrasies in the support set, resulting in a reduced performance when faced with new, unseen instances [71].

### 7.9. Deep Reinforcement Learning

Deep reinforcement learning (DRL) is a type of machine learning that enables machines to learn and make decisions under complex environments. It integrates the power of DNNs with the ability of reinforcement learning algorithms to learn through trial and error and to take actions that maximise a cumulative reward signal. The environment provides the agent with observations and rewards based on its actions (Figure 13). The agent is a learning entity that processes received observations through DNNs and selects actions based on the learned policy. DRL has been successfully applied in various domains, including in medicine. It also holds promise in other areas, such as medical diagnosis and decision making, clinical trial design, and robotic surgery.

### 7.10. Transformer Neural Networks

Transformer neural networks (TNN), also known as transformers, are powerful neural networks that have been widely used in natural language processing [36]. Transformers were developed to solve the problems of sequence-to-sequence transduction and neural machine translation. This includes speech recognition, text-to-speech transformation, etc. They consist of encoder–decoder layers and are trained through pre-training and fine-tuning. (Figure 14) Unlike the traditional recurrent neural networks (RNNs), which process inputs sequentially, transformers exploit a self-attention mechanism (also known as scaled dot-product attention) to weigh the importance of different elements in a sequence, enabling parallelisation and capturing long-range dependencies more effectively. In doing so, the model can thereby assign higher weights to relevant words or tokens and pay attention to them more effectively during processing. While transformers have been highly successful in natural language processing, they also have limitations. These include a high computational complexity, memory requirements for long sequences, a lack of explicit sequential order modelling, limited interpretability, data-intensive training, difficulty with contextual understanding, and challenges with out-of-vocabulary words.

### 7.11. Attention Mechanism

The attention mechanism is a component of artificial intelligence (AI) models that enables them to focus on specific parts of input data that are deemed relevant for a given task. It allows the model to assign weights to the different elements of the input data, directing its attention to the most important information (Figure 15) [75].

In medical imaging, attention mechanisms can be used to highlight specific regions or structures within an image. For instance, in a chest X-ray, attention can be directed to areas of potential pathology, thereby assisting radiologists in their diagnosis and improving the accuracy of automated image analysis algorithms.

Attention mechanisms can aid in clinical decision-making by directing the model’s attention to the relevant information within patient records. When analyzing electronic health records (EHRs) or the medical literature, the model can focus on crucial clinical features, symptoms, or treatment options, thereby assisting healthcare professionals in making informed decisions [76].

The weakness of these attention mechanisms in AI is that they can be sensitive to input variations, resulting in the formation of different attention distributions for similar inputs. This sensitivity can lead to an instability and inconsistency in the model’s attention patterns, thereby affecting its performance. Additionally, attention mechanisms may struggle to handle long-range dependencies in sequences, making it challenging to capture and incorporate information from the distant parts of the input sequence more effectively.

## 8. Examples of AI Applications in Medicine Approved by the US Food and Drug Administration (FDA)

Apple IRNF 2.0: Apple Inc. developed the IRNF 2.0 software for the Apple Watch, utilising CNNs and machine learning algorithms. This software aims to identify cardiac rhythm disorders, particularly atrial fibrillation (AFib). The study conducted by Apple in 2021, with FDA approval, involved over 2500 participants and collected more than 3 million heart rate recordings. The algorithm successfully differentiated between AFib and non-AFib rhythms, with a sensitivity of 88.6% and a specificity of 99.3%. While effective, it is important to note that the app does not replace professional diagnosis nor target individuals that have already been diagnosed with AFib [77].Ultromics: An AI-powered solution detects heart failure with a 90% accuracy, specifically heart failure with preserved ejection fraction (HFpEF). It analyses LV images using an AI ML-based algorithm to accurately measure LV parameters, such as volumes, left ventricular ejection fraction (LVEF), and left ventricular longitudinal strain (LVLS). This software (EchoGo Core 2.0) also classifies echocardiographic views for quality control. AI readings are more consistent than manual readings, regardless of the image quality. Additionally, AI-derived LVEF and LVLS values have been significantly associated with mortality in-hospital and at final follow-up [78].Aidoc: Aidoc is an AI-based platform that permits a fast and accurate analysis of X-rays and CT scans. It detects conditions, like strokes, fractures, and cancerous lesions. Their advanced AI softwares (Aidoc software from 2.0 and above) help radiologists to prioritise their critical cases and expedite patient care. Aidoc has 12 FDA-approved tools, including softwares (Aidoc software from 2.0 and above) for analysing head CT images (detecting intracranial haemorrhage), chest CT studies (identifying aortic dissection), chest X-rays (flagging a suspected pneumothorax), and abdominal CT images (indicating suspected intra-abdominal free gas) [79].Riverain Technologies: This AI-based platform enables an accurate and efficient analysis of lung images for detecting and monitoring lung conditions, like cancer, COPD, and bronchial disease. It includes features such as CT Detect for measuring areas of interest, ClearRead CT Compare for comparing nodules across studies, and ClearRead CT Vessel Suppress for enhancing nodule visibility. This patented technology improves the accuracy and reading performance, and seamlessly integrates processed series with the original CT series for synchronised scrolling [80].

## 9. Discussion and Limitations

The development of artificial intelligence faces many challenges, and a lack of access to the relevant training data is a major problem in training AI models. Manual data labelling is costly, time consuming, and error prone [81]. It is important to ensure the reliability and interpretability of AI and data fusion methods, requiring regulation and ethical considerations. Research on the quality of healthcare services, privacy, and limitations in applying AI to healthcare is crucial [82]. With the development of artificial intelligence, the risk of hacking attacks on medical data, including patient data, is increasing. Data protection in the medical sector is becoming extremely significant, requiring advanced solutions and close cooperation between security specialists, medical IT specialists, and healthcare professionals. The use of AI to monitor and detect anomalies in IT networks and the continuous improvement of security features are key for minimizing the risk of hacking attacks and for protecting the patient data. An essential part of protecting patient data is the development of intelligent systems that use artificial intelligence to monitor and detect anomalies in IT networks. Artificial neural networks optimised for game theory can be used to detect anomalies and identify potential threats in real time [83].

Although machine learning is widely used, there are also some negative sides and limitations of its methods. Firstly, a dependency on large amounts of labelled data, meaning careful data preparation and evaluation, is a time-consuming process [61]. Insufficient data can lead to overfitting or poor generalization, and annotating such data is a challenging process that can lead to the inadequate performance of the model. Acquiring adequate data in medicine is even more challenging, as there are considerable privacy concerns, limited access to medical records, and the need for expert annotations during the learning process [84,85]. Ensuring data security remains a critical challenge, as data breaches, along with an unauthorised access or mishandling of patient data can present legal and ethical consequences, as well as lead to breaches in confidentiality and the erosion of patients trust. Deep neural networks, particularly with multiple hidden layers, which can be considered as black boxes, lack interpretability [86,87]. Understanding how the model arrives at its predictions or explaining the reasoning behind its decisions can be demanding. This lack of interpretability can limit their applicability in domains where transparency is crucial. Furthermore, the model’s generalisation can be questionable, since DNNs can also learn biases present in the training data, which leads to biased predictions and decisions [88]. Additionally, applying DNNs to novel and unseen scenarios remains a significant challenge. The clinical adoption and validation of AI-based systems can be a complex process, as it requires integration with the existing infrastructure, acceptance by healthcare professionals, and validation against previously established standards [89]. On the other hand, there is a risk of overreliance on machine learning models, which can result in the potential neglect of clinical expertise and suboptimal patient care. Healthcare workers can be tempted to heavily rely on model outputs without considering the context and patient characteristics. It is crucial to recognise and address these limitations to ensure the effective application of such systems in medicine.

## 10. Usability

Our study serves as a unique, comprehensive guide, akin to an AI encyclopedia, that explores and explains the concepts of machine learning, artificial neural networks (ANNs), and deep learning (DL). It was created to help other researchers and practitioners understand the concepts of these techniques and provide concrete examples of their applications, highlighting their potential to revolutionise healthcare by improving diagnostics, disease prediction, drug development, and personalised treatment approaches.

Our team of authors strongly believes that our work is a one-of-a-kind and relevant paper. We could not find other papers investigating our topic. Our review contains general information about AI, which was broken down into individual techniques that are used in it to familiarise readers with basic information on the subject. Other articles cover our selected topics in detail, often going into technical details or focusing on only one selected field of medicine. Only one of the papers we identified treated a similar topic, but its aim was different from ours. The review of Ahmad et al. presented a variety of approaches to the practical applications of AI in clinical practice that were extended with a theoretical elaboration that was broader than ours [90]. They provided a large number of examples, which were grouped according to the target specialty. We also provided numerous examples of different models’ usage but lacked the division to certain specialties. In the aforementioned work, only applications of AI methods, in general, were presented—this research was not focused on how the model works and how to distinguish it from other models. However, in our work, the main focus, and a great advantage, was the incorporation of the most important basics along with the presentation of the information about the individual artificial intelligence algorithms with additional diagrams showing how they work supported by examples of their practical applications.

We firmly believe that our work is an excellent resource for beginners, serving an accessible starting point for those new to the field of AI in medicine. Serving as a compendium, it condenses a vast amount of information into a useful guide, making it an invaluable asset for researchers, practitioners, and all of those worldwide who are intrigued by the fusion of AI and medicine.

## 11. Summary and Future Research

Techniques using artificial intelligence present a promising tool employed in medical fields. Observing the speed of the development of these AI methods, it is certain that in the near future, the use of these methods will be an indispensable part of doctors’ work, improving the diagnostic and therapeutic process. Research shows that AI methods are proving themselves not only as tools for identifying numerous diseases, regardless of the diagnosis carried out by specialists, but also as predictive tools for given disease entities, or as tools for observing changes in a patient’s state of health. To date, some of these solutions have been approved by the FDA and have since been implemented for clinical use. Along with the developments above, doctors will be required to take a proactive approach to learning innovative medical technologies, which will undoubtedly make their work easier while improving medicine sectors.

The potential advancements of AI in medicine are vast and transformative. Future research should explore the integration of AI with emerging technologies, address ethical concerns, and focus on developing robust AI models that can enhance diagnosis, for example in X-rays, MRIs, and CT scans [91]. It can also consider factors, such as patient demographics, genetics, and comorbidities to optimise treatment strategies [92]. AI can optimise healthcare workflows by automating administrative tasks, predicting patient flow and hospital resource utilization, and providing real-time decision support for healthcare professionals [93]. Through pursuing these research directions, we can harness the full potential of AI to improve patient outcomes, enhance healthcare efficiency, and drive innovations in the field of medicine. Future research in AI and medicine should also focus on developing interpretable and explainable AI models to enhance trust and transparency. This will enable healthcare professionals to understand and validate the reasoning behind AI-generated recommendations, thereby facilitating collaboration between humans and machines in clinical decision-making. Additionally, exploring the integration of AI with emerging technologies, such as blockchains, the Internet of Things (IoT), and augmented reality (AR), can open up new possibilities for secure and immersive healthcare experiences. Through continuously advancing AI research, we can unlock its full potential to revolutionise healthcare delivery and improve patient care.

## Figures and Tables

**Figure 1 diagnostics-13-02582-f001:**
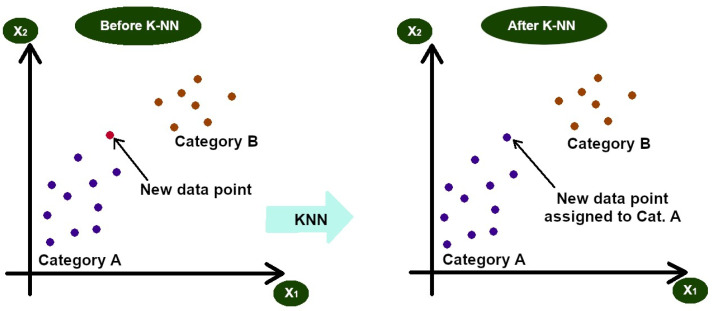
k-nearest neighbour (kNN) example p—new data before and after kNN.

**Figure 2 diagnostics-13-02582-f002:**
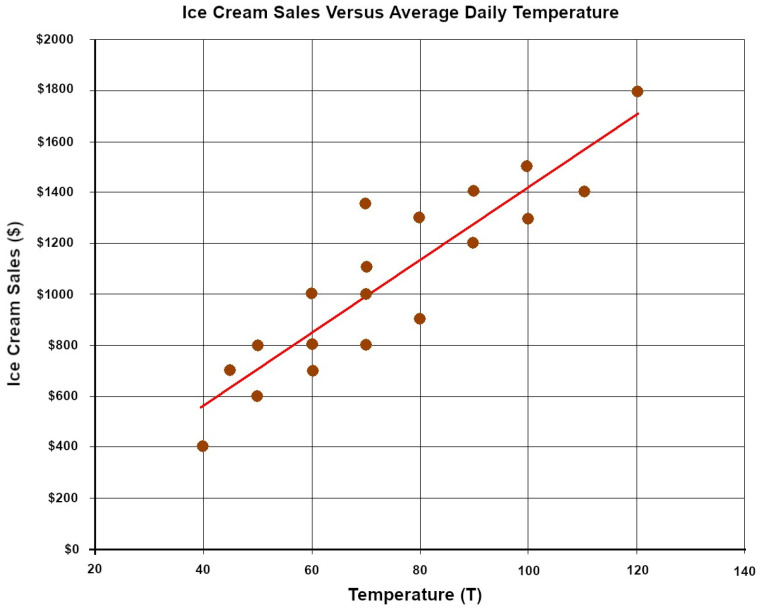
Linear regression example—ice cream sales versus average daily temperature—individual values on subsequent days are represented by brown circles. The red line stands for the linear regression plot created from this data.

**Figure 3 diagnostics-13-02582-f003:**
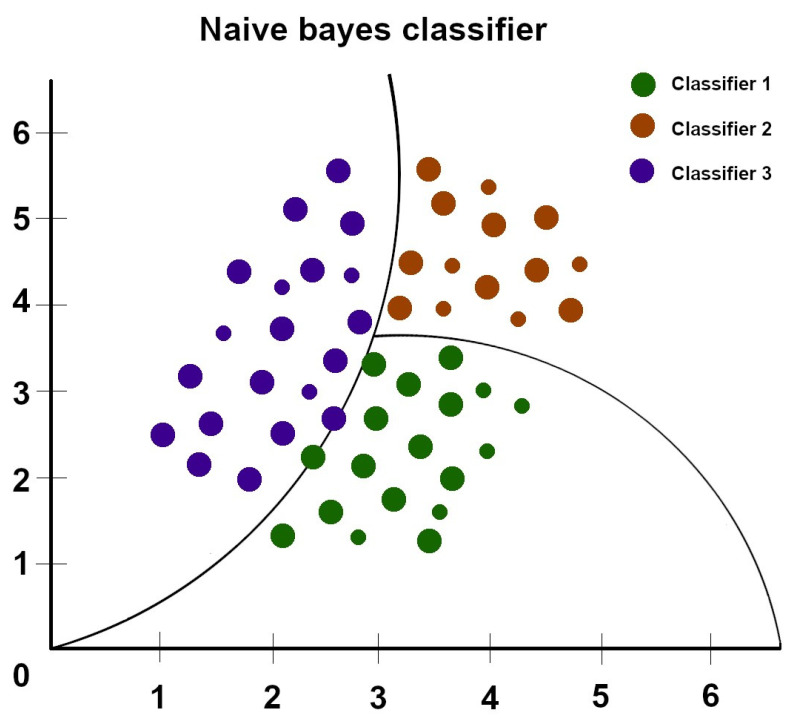
A function graphically depicting the performance of a Naive Bayes classification algorithm.

**Figure 4 diagnostics-13-02582-f004:**
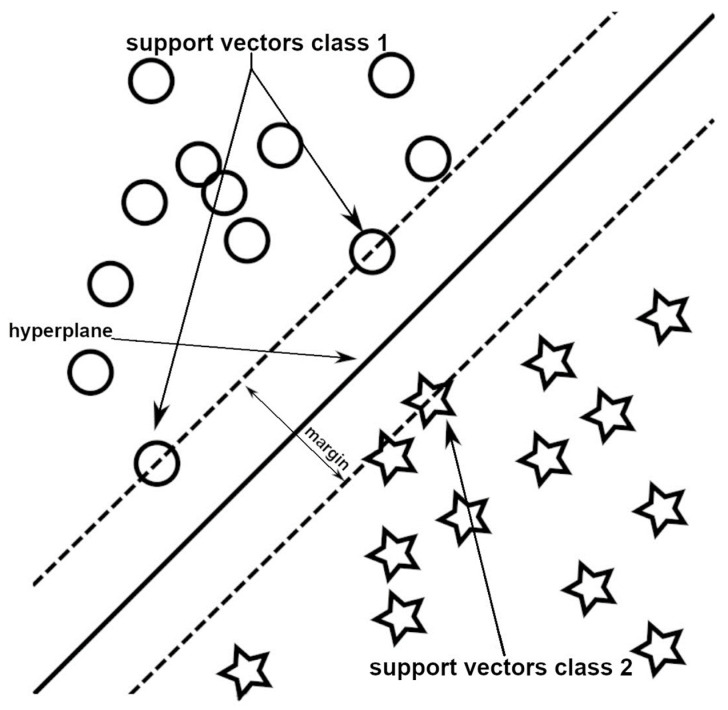
A simplified example of the support vectors and samples of two classes.

**Figure 6 diagnostics-13-02582-f006:**
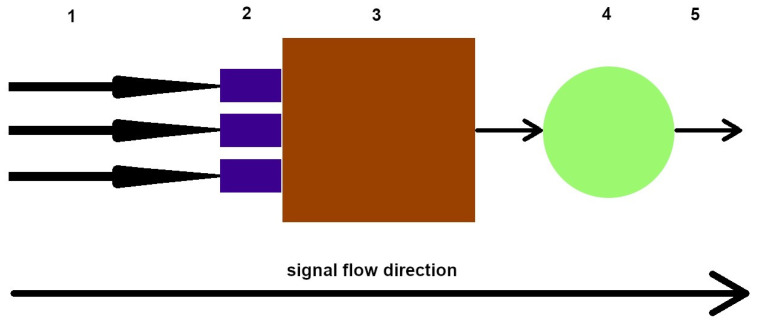
A simplified diagram of a mathematical neuron. 1—signal inputs, 2—scales, 3—adder, 4—activator (activation function), and 5—signal output, respectively.

**Figure 7 diagnostics-13-02582-f007:**
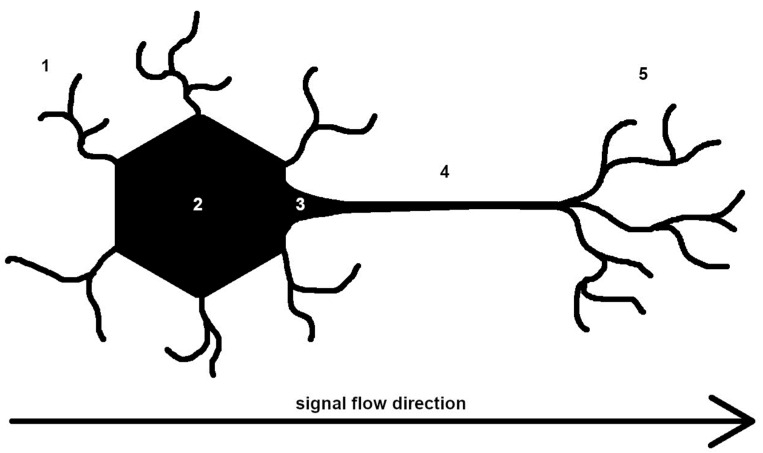
A simplified diagram of a human neuron. 1—dendrites, signal input site, 2—nucleus of the neuron, 3—zone of initiation (where the action potential of the neuron is formed), 4—axon, and 5—axon terminals (which form connections with other cells, and are the sites of signal output), respectively.

**Figure 8 diagnostics-13-02582-f008:**
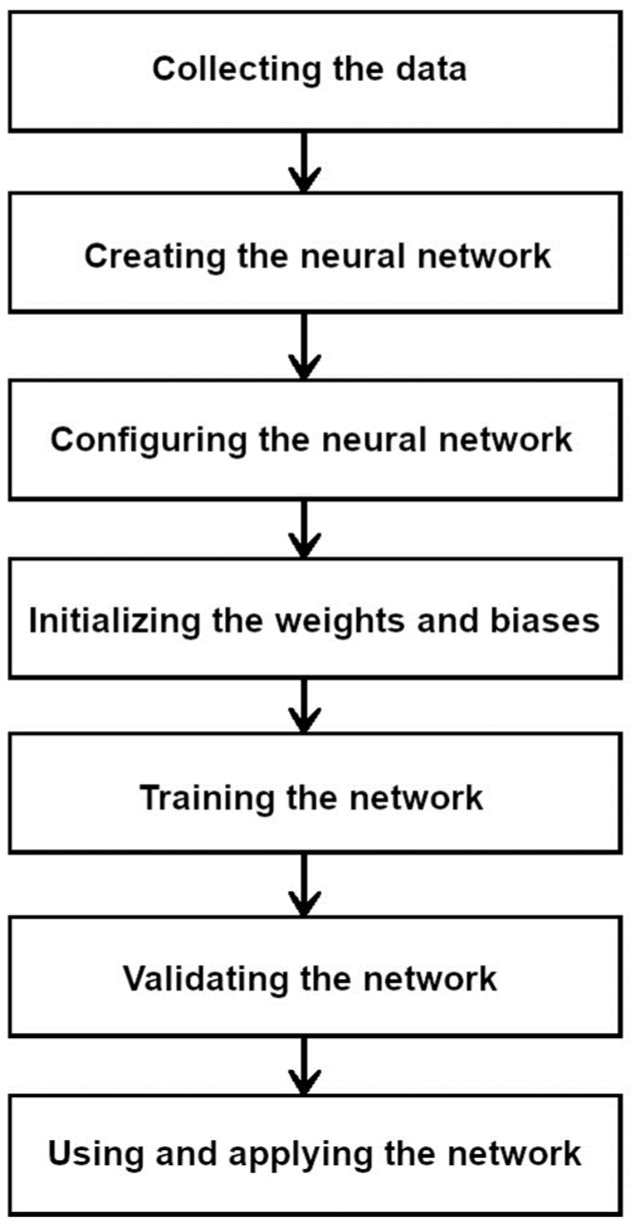
Simplified diagram of the neural network operation.

**Figure 9 diagnostics-13-02582-f009:**
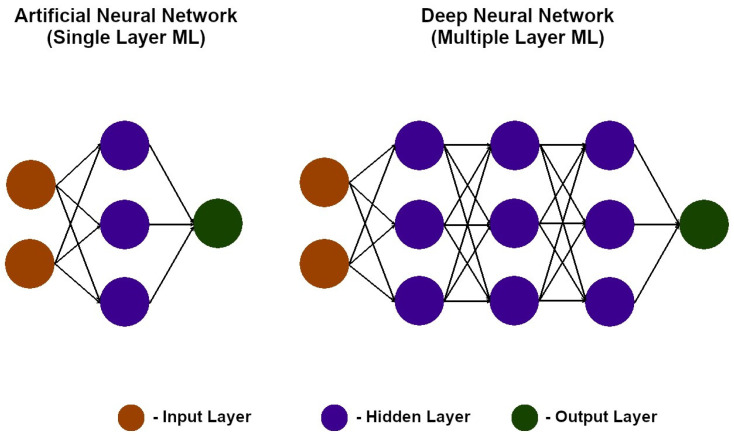
Graphical representation of an artificial neural network (ANN) and a deep neural network (DNN) [39].

**Figure 10 diagnostics-13-02582-f010:**
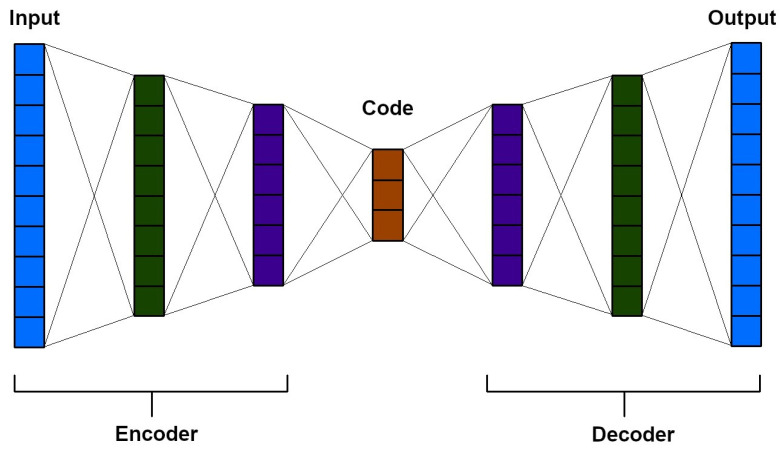
Graphical representation of the auto-encoder [64]. The light blue colour indicates the input and output layers. The dark green colour together with the dark blue colour indicates the internal layers of the encoder and decoder. Red, on the other hand, stands for code.

**Figure 11 diagnostics-13-02582-f011:**
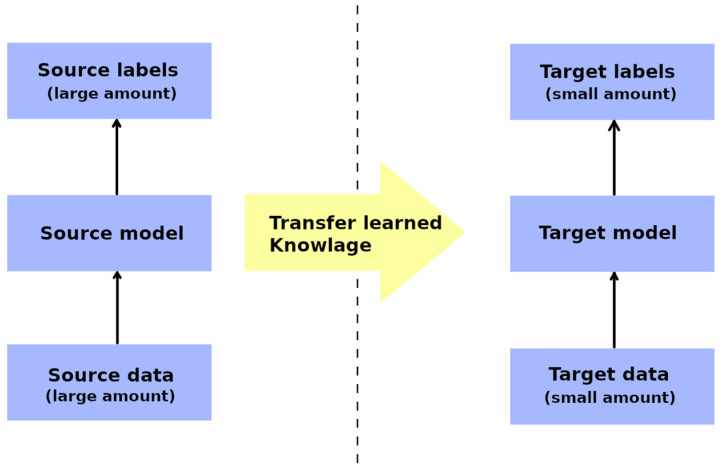
Graphical representation of the transfer learning technique.

**Figure 12 diagnostics-13-02582-f012:**
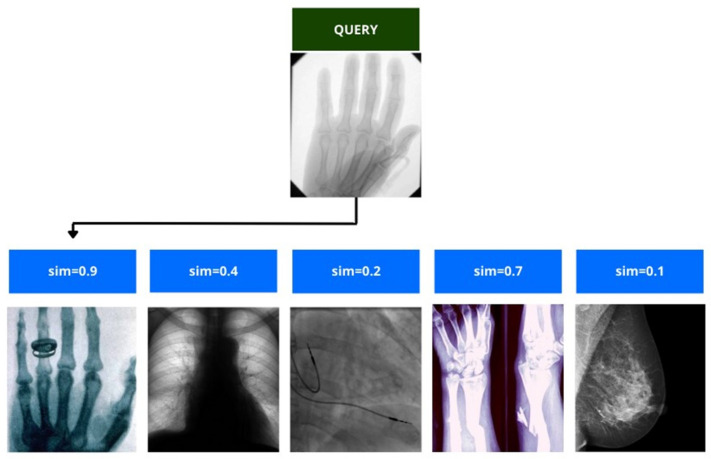
Simplified representation of the few-shot learning paradigm [72].

**Figure 13 diagnostics-13-02582-f013:**
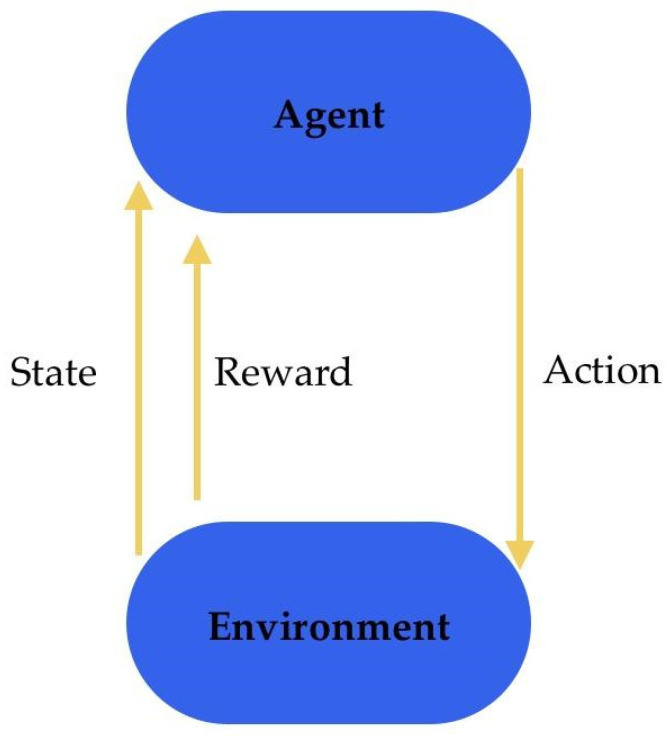
Graphical representation of deep reinforcement learning [73].

**Figure 14 diagnostics-13-02582-f014:**
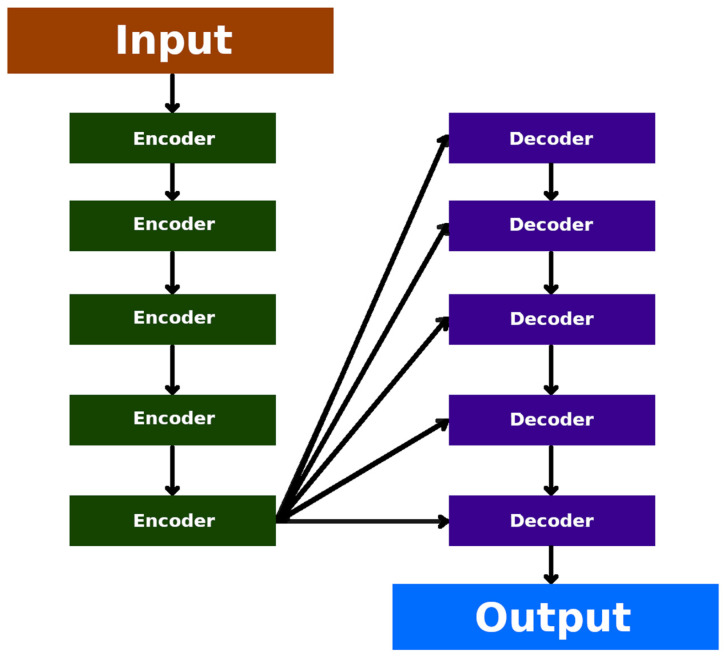
Simplified diagram of TNN operation [74].

**Figure 15 diagnostics-13-02582-f015:**
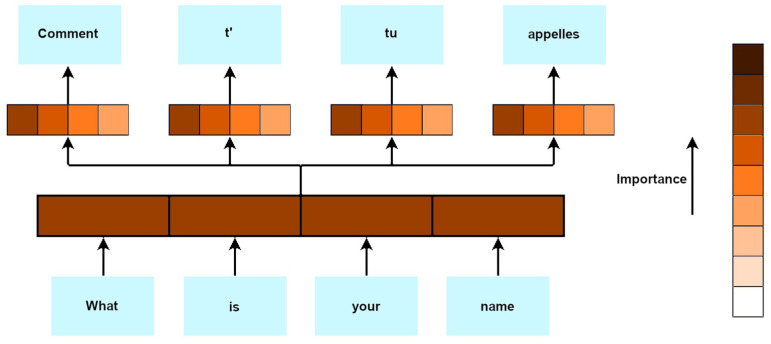
Attention mechanism—simple diagram.

## Data Availability

No new data were created or analysed in this study. Data sharing is not applicable to this article.

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
