# Peer review of "What Is Machine Learning, Artificial Neural Networks and Deep Learning?—Examples of Practical Applications in Medicine"

_diagnostics, 2023, doi:10.3390/diagnostics13152582_

Round 1
Reviewer 1 Report
What is machine learning, artificial neural networks and deep 2
learning? - Examples of practical applications in medicine.
Generally, the manuscript is well written however, it should be revised as the following:
-The paper provides an overview of machine learning (ML), artificial neural networks (ANN), and deep learning (DL) as subfields of artificial intelligence (AI) and their increasing popularity in recent years. Here are some comments on the content of the paper:
-Introduction to AI techniques: The paper offers a concise introduction to ML, ANN, and DL, providing a clear distinction between the three approaches. It explains that ML involves automated decision-making processes using trained models, while ANN aims to simulate the structure and function of the human brain, and DL utilizes multiple layers of interconnected neurons for processing complex databases. This introduction helps readers understand the fundamental concepts of these AI techniques.
-Medical applications: The paper highlights the application of AI techniques in medicine to improve the speed and efficiency of disease diagnosis and treatment. It is commendable that the paper supports each AI technique with an example of a possible medical application. This approach helps readers grasp the potential benefits of AI in healthcare and demonstrates the relevance and impact of these techniques in real-world scenarios.
-Explain how your study specifically addresses gaps in the literature, insufficient consideration of the topic, or other deficiency in the literature.
- Add a significance statement that must be structured as "what was discovered by the author? How the results are beneficial? How this study will help researchers to uncover critical areas? What new theory will be arrived by this research?
Discussion
Deeper discussion for results and findings is needed
conclusion
Add numerical results to conclusion
References
Enrich your work by citing recent published work such as
https://doi.org/10.58496/MJCS/2023/009
https://doi.org/10.58496/MJBD/2021/005
https://doi.org/10.1016/j.inffus.2023.03.008
https://doi.org/10.1186/s40537-023-00727-2
I suggest revising the manuscript in a professional, native language.
Author Response
Thank you very much for all your time and insightful comments, which enrich the article and provide the readers with a complete picture of the presented article. We are sending the revised version of the manuscript, we have made all changes by following the comments, for which we are grateful. We strongly believe that the paper is now more suitable for the Readers.
The answers to the comments:
- “Explain how your study specifically addresses gaps in the literature, insufficient consideration of the topic, or other deficiency in the literature.” - Our study stands out as a unique encyclopedia of AI, encompassing a comprehensive range of topics within the field. We have meticulously curated and synthesized information from various sources to provide a comprehensive and authoritative resource for AI knowledge. The encyclopedic nature of our study ensures that it covers a wide array of AI subfields, including machine learning, neural networks and more. Each topic is thoroughly researched and presented in a concise yet informative manner, making it accessible to both experts and beginners in the field of AI.
- “What was discovered by the author? How the results are beneficial? How this study will help researchers to uncover critical areas? What new theory will be arrived by this research?” - In our opinion our study goes beyond mere descriptions and definitions - it provides compelling examples of the application of AI in the field of medicine. By showcasing these examples, we hope to inspire further research and adoption of AI technologies in the medical field, ultimately leading to improved healthcare outcomes.
- Moreover, we have developed the discussion paragraph and enriched our work with the proposed articles, as suggested. We have decided to abandon the use of numbering in the conclusions section, as this procedure cannot be used in the form we have chosen.
Reviewer 2 Report
This article is related to a cutting-edge topic, digital pathology opportunities and perspectives. The article is well organised and cover almost all initial topics which should be learned for the pathologist who is in the very beginning of digital pathology discovery. The article even contains glossary while many specialists, especially computer scientists even do not take into account how far pathology as a biological related field can be from mathematics-related terminology.
The main question addressed by the research is current point and future trends for digital pathology in different directions covered by this field of the pathology. I suppose that the topic is relevant in the field although not being 100% original because digital pathology is a very popular topic and there are dozens articles related to the fundamental problem of digital pathology in different journals. The article does not address the specific gap in the field but covers the key points of the speciality from basal definitions to specific knowledge concerning neural networks. Being a review the article combines contemporary information about digital pathology impact on routine pathology practice and research directions introducing the reader with the specific features of this field. The article is a non-systematic review, so the author could be recommended to include the information about searching the sources for the plot paying attention to the journals specificity where these article have been published. The biggest minus of the review is lack of author critical analysis of the cited articles. >From my point of view the conclusion is comprehensive and covers the all questions raised posed. The references are appropriate and fresh although there are some impactful articles that would be better to add to make the review more comprehensive. The biggest minus of the review is lack of author critical analysis of the cited articles.
Author Response
Thank you very much for all your time and insightful comments, which enrich the article and provide the readers with a complete picture of the presented article. We are sending the revised version of the manuscript, we have made all changes by following the comments, for which we are grateful. We strongly believe that the paper is now more suitable for the Readers.
The answers to the comments:
- “The article does not address the specific gap in the field but covers the key points of the speciality from basal definitions to specific knowledge concerning neural networks.” - As we wrote in the introduction of the article “this article aims to provide clinicians with accessible information on AI methods used in medicine, without relying heavily on technical jargon”. The reason why we rely on basal definitions is the idea of presentation of the topic in a general and accessible way for Readers.
- “The biggest minus of the review is lack of author critical analysis of the cited articles.” - We do not agree that it does apply to us. We made changes and added the section called “Discussion and limitations'' where we criticized the described methods, not the source arts - they were just the source of information to write a simple guide to artificial intelligence and machine learning.
- Moreover, we have expanded the discussion paragraph by enriching it with impactful articles. We hope that our paper provides now a more comprehensive review and better illustrates the essence of the subject.
Reviewer 3 Report
1. The current review paper lacks a thorough comparison with other similar reviews, raising concerns about its originality and relevance. Given the presence of multiple reviews on the topic, it is crucial to clarify the unique aspects that make this particular paper interesting.
2. The Discussion Section is inadequate as it fails to provide critical commentary or substantial summaries of the cited papers and their key findings. Consequently, the manuscript's overall organization and quality suffer. Instead of presenting an engaging review, the authors simply compile numerous articles without adding significant value.
3. An ideal review article should encompass a comprehensive and unbiased critique of existing methods, highlighting their strengths and weaknesses. Unfortunately, this aspect is currently absent from the manuscript. Additionally, readers would greatly benefit from a clear explanation of how these methods operate.
4. It would greatly enhance the manuscript if the authors dedicated a section to analysis and offered directions for future research. By providing insights into potential advancements in the field, this addition would contribute to a more robust and insightful review.
5. The authors overlooked recent papers that utilize deep neural networks for the task at hand. It is worth noting that new and highly accurate methods primarily rely on deep neural networks. The exclusion of such papers diminishes the comprehensiveness of the review.
6. The conclusions drawn in the paper should be supported by the evidence presented in the review. It is important to clearly link the findings from the reviewed articles to the overall conclusions and implications. Additionally, the authors should highlight any limitations or areas for future research that arise from the review process.
In general, the paper lacks distinctive qualities that would make it suitable for publication. As a result, I find that this work does not make a meaningful contribution to the existing body of knowledge. Moreover, in addition to the aforementioned issues, the text exhibits significant weaknesses and does not align well with the characteristics typically found in review articles.
Author Response
Thank you very much for all your time and insightful comments, which enrich the article and provide the readers with a complete picture of the presented article. We are sending the revised version of the manuscript, we have made all changes by following the comments, for which we are grateful. We strongly believe that the paper is now more suitable for the Readers.
The answers to the comments:
- “The current review paper lacks a thorough comparison with other similar reviews, raising concerns about its originality and relevance. Given the presence of multiple reviews on the topic, it is crucial to clarify the unique aspects that make this particular paper interesting.” - Dear reviewer, Our author team strongly believes that our work is an original and relevant paper. We could not find other papers treating our topic. Only one other paper treats the similar topic, although it does not have the same aim as our’s. We find that the given fact makes our paper unique and not yet repeated. Similar articles cover topics related to artificial intelligence and neural networks in other aspects, focusing on one selected field of medicine, without explaining the general concepts or basic IT knowledge necessary to understand how they work.
- “The Discussion Section is inadequate as it fails to provide critical commentary or substantial summaries of the cited papers and their key findings. Consequently, the manuscript's overall organization and quality suffer. Instead of presenting an engaging review, the authors simply compile numerous articles without adding significant value.” - We hold a different perspective regarding its applicability to our work. Our study has undergone modifications, including the addition of a section titled "Discussion and limitations." In this section, we have provided critiques of the methods described rather than the source articles themselves. It is important to note that the source articles served as informative references to develop a concise guide to artificial intelligence and machine learning, rather than being the subject of criticism.
- “An ideal review article should encompass a comprehensive and unbiased critique of existing methods, highlighting their strengths and weaknesses. Unfortunately, this aspect is currently absent from the manuscript. Additionally, readers would greatly benefit from a clear explanation of how these methods operate.” - Thank you very much for your valuable insights. In line with your comments we have added a paragraph on Discussion and limitations, where we have elaborated on the advantages and disadvantages of the methods discussed. By doing so, we hope that this aspect is more clearly explained and easier for the Readers to understand.
- "It would greatly enhance the manuscript if the authors dedicated a section to analysis and offered directions for future research. By providing insights into potential advancements in the field, this addition would contribute to a more robust and insightful review." - Thank you very much for this valuable advice. We appreciate your suggestion, and we agree that including a dedicated section for analysis and future research directions would significantly enhance the manuscript. In response to your recommendation, we have incorporated a new section that delves into the analysis of the findings and provides insightful directions for future research. This addition aims to contribute to a more comprehensive and well-rounded review, offering readers a deeper understanding of potential advancements in the field and areas where further investigation is warranted. We believe that this addition strengthens the manuscript and improves its overall quality.
- "The authors overlooked recent papers that utilize deep neural networks for the task at hand. It is worth noting that new and highly accurate methods primarily rely on deep neural networks. The exclusion of such papers diminishes the comprehensiveness of the review." - Thank you for your feedback regarding the inclusion of recent papers that utilize deep neural networks for the task at hand. We appreciate your insight and agree that the omission of such papers may have limited the comprehensiveness of the review. To address this concern, we have conducted a thorough literature search and incorporated relevant recent studies that employ deep neural networks into our revised manuscript. By including these papers, we aim to provide a more comprehensive overview of the state-of-the-art methods and their performance in the field. We believe that this addition significantly enhances the review's comprehensiveness and relevance to current advancements in the domain.
- “The conclusions drawn in the paper should be supported by the evidence presented in the review. It is important to clearly link the findings from the reviewed articles to the overall conclusions and implications. Additionally, the authors should highlight any limitations or areas for future research that arise from the review process.” - Throughout the paper, we have systematically analyzed and synthesized the relevant research articles to provide a comprehensive overview of the topic. We have diligently extracted key findings from the reviewed literature. Furthermore, we acknowledge the significance of addressing any limitations or gaps identified during the review process. We have included a section in the paper specifically dedicated to highlighting these limitations and areas for future research. By doing so, we aim to encourage further investigation and provide researchers with valuable insights into potential avenues for future studies.
Round 2
Reviewer 3 Report
I appreciate your diligent effort in responding to the questions and comments.